# Infection as an Important Factor in Medication-Related Osteonecrosis of the Jaw (MRONJ)

**DOI:** 10.3390/medicina57050463

**Published:** 2021-05-09

**Authors:** Sven Otto, Suad Aljohani, Riham Fliefel, Sara Ecke, Oliver Ristow, Egon Burian, Matthias Troeltzsch, Christoph Pautke, Michael Ehrenfeld

**Affiliations:** 1Head of Department of Oral and Maxillofacial Surgery, Martin-Luther-University Halle-Wittenberg, 06120 Halle (Saale), Germany; 2Department of Oral and Maxillofacial Surgery and Facial Plastic Surgery, Ludwig-Maximilians-University, 80337 Munich, Germany; riham.fliefel@med.uni-muenchen.de (R.F.); sara.ecke@med.uni-muenchen.de (S.E.); Matthias.Troeltzsch@med.uni-muenchen.de (M.T.); christoph.pautke@gmx.de (C.P.); michael.ehrenfeld@med.uni-muenchen.de (M.E.); 3Department of Oral Diagnostic Sciences, Division of Oral Medicine, King Abdulaziz University, 80200 Jeddah, Saudi Arabia; suadaljohani@yahoo.com; 4Experimental Surgery and Regenerative Medicine (ExperiMed), Department of Orthopaedic Surgery, Ludwig-Maximilians-University, 80337 Munich, Germany; 5Department of Oral and Maxillofacial Surgery, Alexandria University, Alexandria 21514, Egypt; 6Department of Oral and Maxillofacial Surgery, University of Heidelberg, 69120 Heidelberg, Germany; oliver.ristow@med.uni-heidelberg.de; 7Department of Neuroradiology, Technical University of Munich, 81675 Munich, Germany; Egon.Burian@mri.tum.de

**Keywords:** medication-related osteonecrosis of the jaw, MRONJ, bisphosphonates, denosumab, infection, periodontitis

## Abstract

Medication-related osteonecrosis of the jaw (MRONJ) has become a well-known side effect of antiresorptive, and antiangiogenic drugs commonly used in cancer management. Despite a considerable amount of literature addressing MRONJ, it is still widely accepted that the underlying pathomechanism of MRONJ is unclear. However, several clinical and preclinical studies indicate that infection seems to have a major role in the pathogenesis of MRONJ. Although there is no conclusive evidence for the infection hypothesis yet, available data have shown a robust association between local infection and MRONJ development. This observation is very critical in order to implement policies to reduce the risk of MRONJ in patients under antiresorptive drugs. This critical review was conducted to collect the most reliable evidence regarding the link between local infection and MRONJ pathogenesis.

## 1. Introduction

Bisphosphonates are stable analogues of inorganic pyrophosphate (PPi) which bind to hydroxyapatite crystals at sites of active bone remodeling. They impair intracellular signaling in osteoclasts interfering with osteoclast-mediated bone resorption and therefore are considered one of the most effective antiresorptive drugs (ARDs). Despite the great benefits of bisphosphonates in management of bone metastasis and osteoporosis, a rare but serious side effect, osteonecrosis of the jaw, was reported in 2003. Since then, the number of reported cases has increased dramatically to the point that a causal link has been established between osteonecrosis of the jaw and bisphosphonates intake and the condition was then named bisphosphonate-related osteonecrosis of the jaw (BRONJ). Maxillofacial surgeons observed this complication and thus, a position paper was developed by the American Association of Maxillofacial Surgery (AAOMS) in 2007 to set strategies for treatment and prevention of BRONJ [1,2].

In 2010, osteonecrosis of the jaw was reported in association with the new antiresorptive, denosumab, a monoclonal antibody against the receptor activator of nuclear factor-κB ligand (RANKL) [3]. The same complication was also observed after administration of antiangiogenic medications and tyrosine kinase inhibitors, however there is still little scientific evidence to confirm the association between these medications and osteonecrosis of the jaw [4]. Based on the increasing number of medications that cause osteonecrosis of the jaw, AAOMS proposed the name medication-related osteonecrosis of the jaw (MRONJ) in its last position paper in 2014 in which MRONJ was defined as exposed bone in the jaws or the maxillofacial region that persisted for a minimum period of two months in a patient who has a history of current or previous ARDs or antiangiogenic agents in absence of radiotherapy or metastasis to the jaw [5]. However, recently, the definition of MRONJ has been updated being not only due to the intake of bisphosphonate drugs (current or past), but also further pharmacological therapies such as other antiresorptive agents, or drugs with anti-angiogenic activity. It is also important to perform a thorough physical examination and medical history, together with targeted radiologic examinations. Furthermore, it is important to take into the account not only the presence of exposed necrotic bone but consider also other clinical signs and first/second-level imaging and consider that pain may not always be present in MRONJ cases, especially in the early stages as well as considering that some cases of MRONJ can arise from the presence of dental–periodontal diseases or spontaneously, without any relation to invasive dental procedures [6].

Despite the enormous research efforts made in relation to MRONJ, its pathogenesis is still not fully elucidated. Many hypotheses have been postulated, such as suppression of bone remodeling, inhibition of angiogenesis, constant microtraumas and local infection. However, none of them can fully explain the exact mechanism of MRONJ development and, generally, the process seems to be multifactorial [5]. Clinical observations support ARDs type, dose and frequency as absolute risk factors for MRONJ [5]. Nevertheless, MRONJ does not occur in all patients under ARDs and it is clear that other factors are involved in its onset. Indeed, a considerable and growing body of evidence has accumulated over the last few years and suggests a substantial role of local infection of the jaw bones in initiation of MRONJ. Management of MRONJ can be quite complex and challenging and it is totally agreed that prevention is the best way to face MRONJ, especially in cancer patients under long-term antiresorptive therapy. For this reason, it is crucial to develop a clear understanding of the exact pathophysiology of MRONJ to aid in implementing preventive measures before and after ARDs administration. Therefore, the aim of this critical review is to elucidate the current evidence on the impact of local infection on MRONJ pathogenesis in light of the available clinical and experimental studies.

### 1.1. Clinical Presentation

The clinical hallmark of Medication-related osteonecrosis of the jaw (MRONJ) is exposed necrotic jawbone that does not resolve over 8 weeks in a patient taking antiresorptive agents and who has not had radiotherapy of the jaw [7]. Other main clinical manifestations of MRONJ include typical signs of infection such as pain, swelling, redness, fistula formation, pus exudation and abscess formation [8]. Impairment of inferior alveolar nerve function is a rare but typical clinical sign, which is also a well-established sign of osteomyelitis [9]. Moreover, typical signs of advanced stages of MRONJ such as extraoral fistula formation and involvement of the maxillary sinus further support the integral role of infection in the disease process [5].

### 1.2. Location

Around two thirds of MRONJ occurs in the mandible while the maxilla is affected in only one third of the cases with a predilection to the premolar–molar region. The predilection for the mandible is generally attributed to its limited end-arterial blood supply in addition to its higher ratio of cortical to cancellous bone [8]. Therefore, it is not surprising that the mandible is more prone to infection, as in osteomyelitis of the jaw that also predominantly occurs in the mandible. [10]. An obvious explanation for this localization is the higher prevalence of local dental infections, namely apical and marginal periodontitis, in the respective regions. This could be attributed to the fact that root surface of molars is larger in comparison to other teeth and molars/premolars possess root furcations, which are more difficult to accessed with oral hygiene measures. It is also worth mentioning that MRONJ lesions usually originate from the alveolar process, which is frequently affected by odontogenic infections that penetrate into it through root tips foramina and accessory canals or through infected periodontal tissues.

### 1.3. Risk Factors

Numerous potential risk factors for MRONJ have been discussed. However, evidence is still sparse due to the lack of well-controlled prospective studies. Conceivable risk factors can be categorized into three groups: the type and dose of antiresorptives or antiangiogenic drugs, systemic risk factors and local risk factors.

The dosage of antiresorptive drugs is an important risk factor. The oncological dosing of intravenous bisphosphonates (e.g., zoledronate 4mg/month intravenously) and denosumab (120mg/month subcutaneously) are related to a greater risk of MRONJ when compared to dosing schemes used in osteoporosis treatment [1,11]. Consequently, patients suffering from malignant diseases are more often affected with MRONJ. In these patients, additional antiangiogenic and immunosuppressive medications may further contribute to the higher MRONJ incidence. 

For bisphosphonates, the intravenous route of administration can be regarded as a proven risk factor compared to oral bisphosphonates [12]. The cumulative dose, which can be determined by the duration and frequency of administration, is an additional risk factor [13]. In a combined analysis of three prospective trials comparing the efficacy and safety of denosumab with zoledronate in treatment of metastatic bone disease, the incidence of MRONJ increased from 0.5% and 0.8% in the first year to 1.3% and 1.8% in the third year of denosumab and zoledronate intake, respectively [14].

Co-medications and habits represent additional risk factors. Corticosteroid therapy, diabetes mellitus, chemotherapy and smoking are among the most commonly reported potential risk factors for developing MRONJ [15]. While a direct cause-and-effect relationship between these factors and MRONJ remains to be proven. It is remarkable that all of the aforementioned factors can adversely affect the immune system and increase the susceptibility to infection [16]. This fact may be viewed as a further support for the infection hypothesis. 

With respect to local risk factors, poor oral hygiene, pressure sores from ill-fitting prostheses, dental and periodontal diseases and tooth extractions are clearly linked to local infections and thus to MRONJ [17]. A case–control study showed that periodontitis and bone loss were more common in MRONJ patients than in the controls [18]. These results indicate that local infection could be a possible risk, or even causative, factor in the development of MRONJ. Sedghizadeh et al. suggested that extraction and invasive surgical procedures can ease the access of bacteria from the oral cavity into the jaw bones and hence can induce MRONJ [19]. Underlying infection can lead to extraction; this might explain the high frequency of local surgical procedures prior to the onset of the reported MRONJ cases. In a recent systematic review, tooth extraction was the preceding dental event in 61.7% of the cases [11]. Nevertheless, it is likely that the underlying local infection, which is the typical indication for dental extraction, is the actual key factor in development of bone necrosis rather than the extraction itself [20]. Local risk factors are of special importance as they can be influenced by the patient and by adequate preventive dental care [4]. Of note, most of the factors as poor oral hygiene, periodontitis and tooth extraction that can lead to local infections could be prevented [15]. 

### 1.4. Microbiology

Oral biofilm is known to harbor hundreds of bacterial species [21]. Due to the potential causal relation of local infection in MRONJ pathogenesis, many studies aimed to investigate the microbial populations in MRONJ lesions. Several reports linked MRONJ with certain bacterial species, mainly Actinomyces [22]. In fact, Actinomyces species are regularly found in exposed jaw bones, as in osteoradionecrosis and osteomyelitis, independent of previous antiresorptive treatment [23].

Hallmer et al. explored the relation between oral flora and MRONJ using 16S rRNA pyrosequencing techniques [24]. In that study, anaerobic bacteria representative of periodontal microflora, mainly Porphyromonas, Lactobacillus, Tannerella, Prevotella, Actinomyces, Treponema, Streptococcus and Fusobacterium were frequently detected. Thus, it is likely that periodontitis has a great impact on the initiation of MRONJ.

Furthermore, it has been shown that bisphosphonates incorporated in hydroxyapatite can increase the adhesion of different bacterial species and promote biofilm formation in vitro. Kos et al. has evaluated the adhesion of different strains of Staphylococci and Pseudomonas to the hydroxyapatites with and without pamidronate [25]. It was found that the adhesion of Staphylococci on the hydroxyapatite discs coated with pamidronate was seven times more than that of the uncoated discs. Moreover, the adhesion of Pseudomonas was three times more than in the controls. This means that bone loaded with bisphosphonates is more susceptible to infection not only because of the suppression of defense mechanisms, especially osteoclast activity and bone remodeling, but also because bone loaded with bisphosphonates is more prone to bacterial colonization.

### 1.5. Medication Related Osteonecrosis in Other Locations

Osteonecrosis lesions do not exclusively occur in the jaw bones. Several recent publications confirmed the occurrence of osteonecrosis of the ear canal in patients receiving antiresorptive drugs [26,27]. An infectious genesis has been also discussed with regard to these cases as minor trauma to the thin integumental layer that covers the bone and bacterial contamination could lead to an infection resulting in osteonecrosis [27].

### 1.6. Staging

The American Association of Oral and Maxillofacial Surgeons (AAOMS) proposed a system for classifying and staging MRONJ in 2009 [28]. In contrast to the AAOMS position paper 2009, which assigns patients to different stages of disease on the basis, a study by Bedogni et al. [29] had set up a combined clinical and radiological staging system with the aim of pooling BRONJ patients in different groups based on the radiological extent of the disease. The exposed and necrotic bone is the most important feature for the initial diagnosis of MRONJ. However, in the advanced stage, an imaging-based diagnosis is very important as it indicates the absence or presence of bone change as well as its extent [30]. In 2014, the American Association of Oral and Maxillofacial Surgeons (AAOMS) proposed an updates system for classifying and staging MRONJ, which has been generally used subsequently [6]. 

Patients with no apparent necrotic bone are considered to be ‘at-risk’ if they have been treated with bone-modifying agents. Stage 0 patients have no clinical evidence of necrotic bone but present with non-specific symptoms or clinical and radiographic findings. Patients with exposed and necrotic bone or fistulas that probe to the bone and who are asymptomatic and have no evidence of infection are considered as stage 1 MRONJ while patients with exposed and necrotic bone and who have pain and clinical evidence of infection are considered as stage 2. An advanced stage, which is stage 3, is defined as patients with exposed and necrotic bone or fistulas that probe to bone with evidence of infection and at least one defined characteristic [31].

### 1.7. Imaging

Supporting the hypothesis that MRONJ is a disease entity originating from local infection, several imaging findings can be detected in early and advanced stages that resemble characteristics of osteomyelitis. Initially MRONJ and osteomyelitis start with a similar partially positive bone balance and develop similar structural patterns such as mixed radiopaque and radiolucent areas in form of sclerotic and necrotic bone of the mandible and the maxilla which can be visualized in different imaging modalities like panoramic radiographs, CT and MRI [32,33,34]. The clinical gold standard for initial severity assessment is conventional dental radiography. Panoramic radiography is the most widely used modality with the best accessibility. Although, subtle changes in bony structure cannot be displayed adequately [35].

In advanced phases of MRONJ, osteomyelitis and also of osteoradionecrosis, aggravation of the described pathophysiological changes can lead to persisting alveoli, particularly in the mandibular premolar and molar area, localized osteolysis, formation of sequestra and in severe cases to pathological fractures. These complex structural changes can only be visualized anatomically correct in 3D imaging modalities like CT and CBCT [36]. 

Furthermore, bone scintigraphy and PET/CT show comparable results with an increased uptake and metabolism in early stages of MRONJ and osteomyelitis [37]. The diagnostic significance of functional MRI protocols and Molecular Imaging are still to be evaluated, since until today the microscopic changes preceding the described, qualitative macroscopic alterations cannot be displayed and quantified precisely. In the course of recent advances in functional imaging, DCE-MRI can detect dose dependent alterations in bone vasculature after chemo-radiotherapy, thereby providing surrogate markers for bone physiology and pathophysiology leading to sclerosis and necrosis [38]. However, it is important to mention that no imaging modality is able to determinate the definite extent or the exact borders of neither osteomyelitis nor osteonecrosis yet. 

### 1.8. Animal Studies

Several animal studies further support the triggering role of infection in the onset of MRONJ. Abtahi et al. found that MRONJ did not occur in lack of bone exposure in rats that received nitrogen-containing bisphosphonates [39]. The authors concluded that sterile necrosis is not a prerequisite for MRONJ onset. Considering the reduced regenerative capacity of the bone caused by antiresorptive drugs, local infection can lead to accumulation of tissue damage and subsequently necrosis. Moreover, MRONJ did not develop after dental extraction preceded by antibiotic prophylaxis and followed by mucoperiosteal closure in mice receiving bisphosphonates [40]

Another study in rats showed that high dose of zoledronate both exacerbated the inflammatory response and damage to periodontal tissue and induced bone lesions similar to osteonecrosis of the jaw [41].

Tsurushima et al. showed that osteonecrosis developed in rats after inflammatory stimulus not only in the jaw, but also in other bones [42]. In this study, zoledronate was administered subcutaneously for 4 weeks. Subsequently, the rats were divided into three subgroups. In the group of rats that were injected with Aggregatibacter actinomycetemcomitans and CFA in the lower jaw and femur, significantly histologically wider osteonecrosis areas in both the jaw bones and femur were detected than in the group with the applied saline solution. In this rat model, it was shown that bacterial infection is a trigger for osteonecrosis.

The role of infection in the development of MRONJ has been further supported not only in small-animal models but also in a recent large animal minipig model. In this minipig model, infection of extraction sockets led to clinical, radiographic and histological signs of MRONJ at all extraction sites after administration of zoledronate. Another notable observation in this study was that MRONJ lesions did not occur only in areas of tooth extraction but also in areas of marginal periodontitis, periapical pathology and areas of food entrapment [43]. These large animal studies also highlight the relation between local infections and the pathogenesis of osteonecrosis [43,44].

### 1.9. Prevention

The National Cancer Institute of Milan reported that the incidence of MRONJ in oncology patients has decreased from 3.2 to 1.3% after the implementation of preventive dental care [45]. To optimize and limit the risk for MRONJ development it is of utmost importance to eliminate infections in the oral cavity before starting antiresorptive drugs, by improving oral hygiene and keeping regular dental care. In addition, it is desirable to finish the adequate dental treatment of local infections and eventually prosthetic adjustment at this stage. All preventive measures before and under antiresorptive therapy with bisphosphonates or denosumab are directed towards reduction in or avoidance of local infections. In some studies, it was demonstrated that preventive dental measures significantly reduce the incidence of MRONJ [46,47]. Traced back to the fact that local infection often precedes the occurrence of MRONJ, further clinical hints indicate a major role for local infections in the pathogenesis of the disease [4,48].

However, for a long time, it was assumed that tooth extractions are the leading cause for the formation of MRONJ [49,50]. Therefore, some international guidelines and expert recommendations advised or still advise obviation of dental surgical procedures if they are not absolutely necessary [5,46]. Nevertheless, it is becoming increasingly clear that the dental surgical procedure itself, when performed correctly according to the standardized provisos, is not the main trigger for the development of MRONJ. Indeed, it is likely that the local infection leading to an extraction presents the key factor for developing MRONJ. In this respect, relevant studies showed that there are already early MRONJ lesions at the time of dental extractions [20,51]. This is supported by recent studies carving out the causal link between periodontitis and MRONJ [52]. A case–control study noted that periodontitis and supporting bone loss were more common in MRONJ patients than in controls [18]. These results might indicate that periodontal infection is an important trigger for MRONJ onset.

Therefore, several author groups suggest that tooth extraction can not only be securely performed, but even is necessary when infections are present to reduce the incidence of MRONJ [20,53]. Indeed, this can be conducted even in high-risk patients taking antiresorptive medications with underlying metastatic malignant disease, when performed in accordance to specified provisos: (i) prolonged preoperative, systemic antibiotic shielding, (ii) atraumatic surgery, (iii) removal of sharp bone edges and the upmost layer of the bones in the sense of modelling osteotomies, (iv) primary plastic saliva-proofed wound closure with a mucoperiosteal flap [54]. Taking a closer look, all those measures are recommended to be performed in order to prevent and remove possible early lesions, to avoid bacterial colonization of the alveolar bone and to prevent subsequent infection, and to thereby reduce the incidence of MRONJ [10,55]. The obvious effectiveness of these measures further proves the key role of infection in the pathogenesis of MRONJ.

### 1.10. Treatment

While there is still no general consensus on the optimal treatment modalities it is remarkable that practically all applied treatment strategies are well known anti-infective strategies. The non-surgical or so-called conservative treatment usually consists of anti-infective and disinfective mouth rinses and long-term antibiotic treatment often accompanied by a temporary, or even permanent, stop of antiresorptive drug treatment. While this treatment often does not lead to complete mucosal healing, it can certainly lead to a symptomatic improvement as it can calm down the signs of infection [56]. It may also lead to a down-staging of the disease (e.g., from stage 2 to stage 1) due to reduction in pain, swelling and pus exudation [57]. It has been recently reported that surgical treatment of the early stages of ONJ (stages 1 and 2) can rapidly improve the clinical conditions of MRONJ patients, achieving complete wound healing without local infection and promoting a better quality-of-life [58]. Another study strongly suggested that MRONJ occurring both in neoplastic and non-neoplastic patients benefits more from a surgical treatment approach, whenever deemed possible, as non-surgical treatments do not seem to allow complete healing of the lesions [59]. Likewise, the surgical treatment consists of perioperative antibiotic therapy, complete removal of necrotic bone parts, smoothening of sharp bony edges and plastic wound closure. All of these measures are directly addressing infection, removing necrotic and infected bone parts and protecting the surrounding bone from reinfection.

Even the potential supportive treatment options including ozone therapy, hyperbaric oxygen, the use of autologous platelet concentrates [60] or low-level laser therapy (LLLT) are well known supportive treatment options for infectious conditions [11].

Taken together basically all the reported treatment modalities of MRONJ clearly imply a role of infection in the disease process.

### 1.11. Implants

The insertion of dental implants during antiresorptive drug treatment had initially been speculated to be a risk factor for developing MRONJ or even to be a trigger event [61]. However, several recent articles clearly pointed out that most of the implants in local relationship to MRONJ sites have been placed well before the onset of the antiresorptive drug treatment in the respective patients which exclude implant insertion as a potential trigger for MRONJ onset in the majority of patients [41,62]. There was rather a relationship between clinical and radiological signs of peri-implantitis and MRONJ [62]. Thus, again, infection, in this case in form of peri-implantitis, could be seen as a decisive trigger for the development of MRONJ.

However, the placement of dental implants is, under normal circumstances, a quite sterile procedure and often accompanied by antibiotic prophylaxis. Respectively, implant insertion is rarely the actual cause for MRONJ manifestation.

## 2. Summary

Many mechanisms have been suggested, including over-suppression of bone remodelling, anti-angiogenic properties or direct tissue toxicity of antiresorptive drugs, hyper-occlusal forces and local infection. All of the proposed mechanisms might play a part; however, local infection of the jawbone seems to play a major role in the pathogenesis of MRONJ [63,64]. Recently, other etiopathogenetic theories have been suggested such as the effect of ARDs on mesenchymal precursors of components of the periodontium and alveolar bone [65]. Indeed, no other bone in the human body is as frequently affected by infections compared to the jawbones: more than 50% of young adults suffer from moderate to severe periodontitis. The risk increases with aging to reach 75% of the population aged 65 years or more [66]. It is well-known that osteoclasts play a key role in the physiologic response to bone infections. Not only the phagocytosis’ capability of cell detritus but also the ability to resorb small necrotic bone fragments make osteoclasts an irreplaceable player in this battle.

Although antiresorptive drugs in particular denosumab and bisphosphonates possess completely different mechanisms of action and pharmacokinetics, both drugs have the same target cell, namely osteoclasts.

It is well-known and proven in different studies that the osteoclast activity is significantly reduced under antiresorptive treatment [67]. Thus, simultaneous presence of local infection and MRONJ is not a coincidence but rather indicates a significant correlation between the two.

The molar area of the mandible is known to be frequent site of dental and periodontal infections. Furthermore, the disease process of MRONJ in dentate patients usually starts by infected tooth bearing areas of alveolar process or areas of dento–alveolar surgeries without adequate preventive measures. In edentulous patients, pressure sores might increase the need for remodeling beyond the capacities of patients treated with antiresorptive drugs. Clinical presentation of MRONJ is similar to osteomyelitis and basically all clinical signs of MRONJ are well-known signs of infection [68,69].

The risk factors, namely diabetes mellitus, smoking, poor oral hygiene, steroid intake, immunosuppression, may contribute to an increased risk of infection. Bacterial colonization of MRONJ tissue samples by different bacterial species also underlines the plausible link to local infection as a triggering event [23,70].

Similarly, MRONJ was reported other than the jaw, in the ear canal, which is also characterized by bacterial colonization and only a very thin epithelial layer covering the bone. This underscores the role of infection in the pathogenesis of MRONJ. Imaging and animal studies have also shown the central role of the infection [43,44,71]. Likewise, prevention of MRONJ aims to prevent infection and thus to eliminate the potential key factor of MRONJ pathogenesis. The decisive role of infection is also evidenced by the drop in incidence of MRONJ in cancer patients who managed to improve their dental hygiene, which thereby prevented potential inflammation and infection [45,72]. Of note, the current treatment of MRONJ is based on controlling existing infection to avoid rapid osteonecrosis progression.

Another aspect is the blood supply of infected bone. There are also other conditions that require stronger remodeling activities and vessel ingrowth especially following surgical procedures.

It is also no coincidence that other drugs which might interfere with host defense and wound healing, especially antiangiogenic drugs, have been recently connected to the development of MRONJ. These drugs (e.g., bevacizumab, Sunitinib, mTOR inhibitors) may act as additional risk factors/co factors for the development of MRONJ in patients receiving antiresorptive drugs [73]. The growing number of studies and models has strengthened the claim that assigns infection as the key trigger in the pathogenesis of MRONJ.

No agreement has been reached in the treatment of MRONJ. Some recommendations focus on the administration of antibiotics, oral antibacterial mouth rinse or surgical debridement or a segmental mandibulectomy and partial maxillectomy with mandibular reconstruction with the fibula flap and covering the exposed areas with tissue flaps. Hyperbaric oxygen (HBO) therapy, fluorescence-guided bone resection, and low-intensity laser therapy have also been studied as therapeutic tools. Other treatment modalities that increase bone wound healing using growth factors had been studied. More recently, teriparatide (N-terminal 34-amino acid recombinant human para-thyroid hormone) has been reported for the medical treatment of MRONJ. Pentoxifylline and a-tocopherol in addition to antimicrobial therapy has been shown to decrease the area of bone exposure and symptoms in MRONJ patients. The use of ozone in combination with antibiotics and surgery for patients with exposed bone lesions has also been the subject of a clinical investigation [11].

The intention of this paper is to stress and reinforce the role of infection in the pathogenesis of MRONJ by collecting pieces of evidence that have been published since the introduction of our hypothesis almost 10 years ago [63]. However, that does by no means disprove any other theory. The infection theory might in fact tie all of the other potentially involved factors and co-factors such as remodeling suppression, inhibition of angiogenesis, inhibition of immunocompetent cells and soft tissue toxicity together. Thus, as detailed above, the infection theory can conclusively explain many of the clinical and radiological features of MRONJ as well as the reasons behind all of the relevant prophylactic and therapeutic measurements.

## 3. Conclusions

MRONJ has been under intensive investigation during the past years; however, there remains a remarkable dearth of knowledge regarding its pathogenesis. Based on a review of the currently available evidence, we conclude that local infection plays a key role in the pathogenesis of MRONJ. All of the clinical and radiological signs of MRONJ and also all of the known preventive and therapeutic measures are directly related to infection. In spite of the fact that none of these observations alone reach causality, together they give strong support for this hypothesis.

## Data Availability

Not applicable.

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
