# Peer review of "Infection as an Important Factor in Medication-Related Osteonecrosis of the Jaw (MRONJ)"

_medicina, 2021, doi:10.3390/medicina57050463_

Round 1

Reviewer 1 Report

The manuscript submitted to Medicina entitled “Infection is the main cause of medication-related osteonecrosis: A trial based on circumstantial evidence” is an original review article which aim to evaluate the link between local infection and medication-related osteonecrosis of the jaw (MRONJ) pathogenesis.

On my opinion the article is interesting, well written, with good English. The prestige of the Authors of this manuscript is worldwide recognized. Anyway, I tried to split hairs.

My main suggestion is to include a brief subsection on the MRONJ staging system. My opinion is that the AAOMS staging system is outdated. In my clinical practice I use the staging system described by Bedogni et al. in 2012 [https://doi.org/10.1111/j.1601-0825.2012.01903.x].

The only oversight I have highlighted is the lack of reference to the use of autologous platelet concentrates in the "prevention" and "treatment" subsections. In this regard, I would like to suggest this recent review of the literature to the authors: https://doi.org/10.1016/j.jcms.2020.01.014 .

Thanks for the opportunity to review this manuscript.

Author Response

The authors would like to thank Reviewer 1 for the effort in reviewing the manuscript. A point-by-point response to the reviewer's comments is provided. The modifications have been written in red in the tracking manuscript. 

On my opinion the article is interesting, well written, with good English. The prestige of the Authors of this manuscript is worldwide recognized. Anyway, I tried to split hairs.

Response: The authors would like to thank the reviewer for this comment. We would try to make our best for modifying the manuscript according to your suggestions.

My main suggestion is to include a brief subsection on the MRONJ staging system. My opinion is that the AAOMS staging system is outdated. In my clinical practice I use the staging system described by Bedogni et al. in 2012 [https://doi.org/10.1111/j.1601-0825.2012.01903.x].

Response: According to the reviewer’s suggestions, a subsection has been added to the manuscript about the staging of MRONJ with the corresponding references.

The only oversight I have highlighted is the lack of reference to the use of autologous platelet concentrates in the "prevention" and "treatment" subsections. In this regard, I would like to suggest this recent review of the literature to the authors: https://doi.org/10.1016/j.jcms.2020.01.014 .

Response: According to the reviewer’s suggestions, the use of autologous platelet concentrates has been added as a treatment modality for medication-related osteonecrosis of the jaw with the corresponding reference.

Reviewer 2 Report

The review entitled "Infection is the main cause of medication-related osteonecrosis: A trial based on circumstantial evidence" submitted to Medicina is an interesting manuscript aimed at highlight and illustrating the influence of odontogenic infection on ONJ onset.

I really enjoyed the content of manuscript and would like to congratulate the authors who are experts in the ONJ topic.

I especially appreciated the completeness of the paper and the stress of the concept that many times dental extraction is a necessary act for the management of the infection already present, probable cause of the onj.
In fact, the authors underline that tooth extraction is often not the primum movens that causes onj, however it could be part of the treatment for a better access to the necrotic bone.

The review comprehensively shows the ONJ topic, from etiopathogenesis to diagnosis, specifying the risk factors and the different types of treatment recently suggested by the literature.
Each part of the text refers to infection as the main cause of onset onj, highlighting how prevention is the best possible "treatment" for the management of ONJ patients, particularly in cancer patients.

After these considerations I would underscore some questions about the manuscript:

- "...in which MRONJ has been defined as exposed 51 bone in the jaws or the maxillofacial region that persisted for a minimum period of two 52 months in a patient who has a history of current or previous ARDs or antiangiogenic 53 agents in absence of radiotherapy or metastasis to the jaw [6]."

As underscored in a recent Conference Report, one of the main clinical issues is concerning the presence of exposed necrotic bone, in the oral cavity, as an essential (conditio sine qua non) sign to diagnose MRONJ.

doi:10.3390/ijerph17165998

One of the good clinical practices could be to take into the account not only the presence of exposed necrotic bone but considering also other clinical signs and first/second-level imaging (confirmed).

- "...it may also lead to a down-staging of the disease (e.g., from stage 2 to stage 1) due to reduction of pain, swelling and pus exudation [53]"

I recommend reporting some interesting recent papers where it has been highlighted how surgical treatment of the early stages of ONJ (stage 1 and 2) can rapidly improve the clinical conditions of MRONJ patients, achieving complete wound healing without local infection and promoting a better quality-of-life.
In particular, it is fundamental to quickly recognize the patient with MRONJ who may benefit from surgical treatment to reduce the risks of progression of the infection, allowing a rapid down-staging of the lesions.

doi: 10.1016/j.joms.2020.05.037
doi: 10.1111/odi.12764

- At the beginning of the discussion part, I suggest that the authors can include other recent etiopathogenetic theories such as the effect of ARDs on mesenchymal precursors of components of the periodontium and alveolar bone.
Although it is widely recognized the effect of ARDs on osteoclast, it could be very interesting for the readers to hear about the effects of these drugs on the human mesenchymal stem cells and the related etiopathogenetic hypothesis.

doi: 10.1177/0963689720948497

I totally agree with the authors on the intention to stress and reinforce the role of infection in the pathogenesis of MRONJ.

Author Response

The authors would like to thank Reviewer 2 for the effort in reviewing the manuscript. Below are the point-by-point response of the authors to the reviewer comment. The modifications have been written in red in the tracking mansucript.

The review entitled "Infection is the main cause of medication-related osteonecrosis: A trial based on circumstantial evidence" submitted to Medicina is an interesting manuscript aimed at highlight and illustrating the influence of odontogenic infection on ONJ onset.

I really enjoyed the content of manuscript and would like to congratulate the authors who are experts in the ONJ topic.

Response: The authors would like to thank the reviewer for this comment. We would try to make our best for modifying the manuscript according to your suggestions.

I especially appreciated the completeness of the paper and the stress of the concept that many times dental extraction is a necessary act for the management of the infection already present, probable cause of the onj.

In fact, the authors underline that tooth extraction is often not the primum movens that causes onj, however it could be part of the treatment for a better access to the necrotic bone.

The review comprehensively shows the ONJ topic, from etiopathogenesis to diagnosis, specifying the risk factors and the different types of treatment recently suggested by the literature.

Each part of the text refers to infection as the main cause of onset onj, highlighting how prevention is the best possible "treatment" for the management of ONJ patients, particularly in cancer patients.

After these considerations I would underscore some questions about the manuscript:

- "...in which MRONJ has been defined as exposed 51 bone in the jaws or the maxillofacial region that persisted for a minimum period of two 52 months in a patient who has a history of current or previous ARDs or antiangiogenic 53 agents in absence of radiotherapy or metastasis to the jaw [6]."

As underscored in a recent Conference Report, one of the main clinical issues is concerning the presence of exposed necrotic bone, in the oral cavity, as an essential (conditio sine qua non) sign to diagnose MRONJ.

doi:10.3390/ijerph17165998

One of the good clinical practices could be to take into the account not only the presence of exposed necrotic bone but considering also other clinical signs and first/second-level imaging (confirmed).

- "...it may also lead to a down-staging of the disease (e.g., from stage 2 to stage 1) due to reduction of pain, swelling and pus exudation [53]"

Response: According to the reviewer’s suggestions, a paragraph has been added about the recent definition of MRONJ with the corresponding reference.

I recommend reporting some interesting recent papers where it has been highlighted how surgical treatment of the early stages of ONJ (stage 1 and 2) can rapidly improve the clinical conditions of MRONJ patients, achieving complete wound healing without local infection and promoting a better quality-of-life.

In particular, it is fundamental to quickly recognize the patient with MRONJ who may benefit from surgical treatment to reduce the risks of progression of the infection, allowing a rapid down-staging of the lesions.

doi: 10.1016/j.joms.2020.05.037

doi: 10.1111/odi.12764

Response: According to the reviewer’s suggestions, a paragraph has been added about the surgical treatment of the early stages of ONJ with the corresponding reference.

- At the beginning of the discussion part, I suggest that the authors can include other recent etiopathogenetic theories such as the effect of ARDs on mesenchymal precursors of components of the periodontium and alveolar bone.

Although it is widely recognized the effect of ARDs on osteoclast, it could be very interesting for the readers to hear about the effects of these drugs on the human mesenchymal stem cells and the related etiopathogenetic hypothesis.

doi: 10.1177/0963689720948497

Response: According to the reviewer’s suggestions, a paragraph has been added about the other recent etiopathogenetic theories with the corresponding reference.

I totally agree with the authors on the intention to stress and reinforce the role of infection in the pathogenesis of MRONJ.

Response: The authors would like to thank the reviewer for this comment.

Round 2

Reviewer 2 Report

I congratulate the authors for their research quality. 
They have really enriched the notions present in the literature on ONJ.
This paper focus the attention on a presumable cause in the pathogenesis of MRONJ.

Author Response

The authors would like to thank Reviewer 2 for the comments. Without your valuable recommendations, we would not have reached here so far. Thank you so much again.